# Overexpression of *ThMYC4E* Enhances Anthocyanin Biosynthesis in Common Wheat

**DOI:** 10.3390/ijms21010137

**Published:** 2019-12-24

**Authors:** Shuo Zhao, Xingyuan Xi, Yuan Zong, Shiming Li, Yun Li, Dong Cao, Baolong Liu

**Affiliations:** 1Qinghai Province Key Laboratory of Crop Molecular Breeding, Northwest Institute of Plateau Biology, Chinese Academy of Sciences, Xining 810008, China; 18234111656@163.com (S.Z.); xixy@nwipb.cas.cn (X.X.); zongyuan@nwipb.cas.cn (Y.Z.); lishimingbgi@163.com (S.L.); caod.08@163.com (Y.L.); 2Key Laboratory of Adaptation and Evolution of Plateau Biota, Northwest Institute of Plateau Biology, Chinese Academy of Sciences, Xining 810008, China; 3College of Life Sciences, University of Chinese Academy of Sciences, Beijing 100049, China; 4The Innovative Academy of Seed Design, Chinese Academy of Sciences, Xining 810008, China

**Keywords:** wheat, anthocyanin biosynthesis, bHLH transcription factor, *ThMYC4E*, overexpression, antioxidant activity

## Abstract

The basic helix-loop helix (bHLH) transcription factor has been inferred to play an important role in blue and purple grain traits in common wheat, but to date, its overexpression has not been reported. In this study, the bHLH transcription factor *ThMYC4E*, the candidate gene controlling the blue grain trait from *Th. Ponticum*, was transferred to the common wheat JW1. The positive transgenic lines displayed higher levels of purple anthocyanin pigments in their grains, leaves and glumes. Stripping the glumes (light treatment) caused white grains to become purple in transgenic lines. RNA-Seq and qRT-PCR analysis demonstrated that the transcript levels of structural genes associated with anthocyanin biosynthesis were higher in transgenic wheat than the wild-type (WT), which indicated that *ThMYC4E* activated anthocyanin biosynthesis in the transgenic lines. Correspondingly, the anthocyanin contents in grains, roots, stems, leaves and glumes of transgenic lines were higher than those in the WT. Metabolome analysis demonstrated that the anthocyanins were composed of cyanidin and delphinidin in the grains of the transgenic lines. Moreover, the transgenic lines showed higher antioxidant activity, in terms of scavenging DPPH radicals, in the ethanol extracts of their grains. The overexpression of *ThMYC4E* sheds light on the traits related to anthocyanin biosynthesis in common wheat and provide a new way to improve anthocyanin content.

## 1. Introduction

Anthocyanins are glycosylated polyphenolic compounds that vary in color from orange and red to purple and blue in flowers, seeds, fruits and vegetative tissues [1,2]. As anthocyanins are water-soluble pigments that are mostly located in cell vacuoles, their hues are influenced by the intravacuolar environment [3,4,5]. Over 600 anthocyanins have been identified in Nature. The most common are the derivatives of six widespread anthocyanidins, namely pelargonidin, cyanidin, delphinidin, peonidin, petunidin and malvidin. Anthocyaninscan protect plants against various biotic and abiotic stresses [6], partially due to their powerful antioxidant properties. In addition, anthocyanin-rich food products have become increasingly popular, due to their attractive colors and suggested benefits to human health [4,7,8]. 

Because they are easily observed, the anthocyanin biosynthesis pathway has been studied at length in model plants. The initial precursor, phenylalanine, is catalyzed stepwise by phenylalanine ammonia lyase (PAL), cinnamate 4-hydroxylase (C4H), 4-coumarate-CoA ligase (4CL), chalcone synthase (CHS), chalconeisomerase (CHI), flavanone 3-hydroxylase (F3H), flavonoid 3-hydroxylase (F3′H) or flavonoid 3′,5′-hydroxylase (F3′5′H), dihydroflavonol 4-reductase (DFR), anthocyanidin synthase (ANS) and O-methyltransferase (OMT) [9,10,11]. The transcription of the structural genes is directly regulated by the transcriptional activation of the MYB-basic helix loop helix (bHLH)-WD40 complex (MBW) that consists of R2R3-MYB, bHLH and WD40 proteins [12,13,14]. Purple leaves or fruits rich in anthocyanin usually arise from the transcriptional activation of MYB transcription factors [8,15,16,17]. Together with MYB, the bHLH transcription factors play active and critical roles in anthocyanin biosynthesis. The high anthocyanin content of many crops results from the raised expression of bHLH transcription factors [14,18,19,20].

To date, anthocyanins have been detected in all terrestrial plants, except one order, the Caryophyllales, which accumulate betalains [21]. Many traits in common wheat (*Triticum aestivum*) are related to the biosynthesis of anthocyanins including purple or blue grain, purple leaf, purple coleoptile, red ligule and red leaf [22]. The common wheat with blue and purple grain were used worldwide to get high anthocyanin wheat [23]. As their presence can easily be visualized, classic genetic studies have summarized the genetic laws for these color traits, along with the genetic loci associated with these and other corresponding characteristics. For some characteristics, the major gene, usually the transcription factor MYB or bHLH, that controls the corresponding color trait has been isolated. For example, *Rc* genes (*Rc-A1*, *Rc-B1*, and *Rc-D1* located on the short arms of chromosomes 7A, 7B and 7D, respectively), which control red coleoptiles, encode MYB transcription factors [24,25,26]. *Pp3* and *Pp1*, major genes controlling purple grain [27,28], encode the bHLH transcription factor *TaMYC1* and the MYB transcription factor *TaMYB7D*, respectively [29,30]. The bHLH transcription factor *ThMYC4E* has been inferred to be the candidate gene (*Ba1*) controlling the blue aleurone trait in *T. aestivum* genotypes carrying *Th. ponticum* introgression [31]. Although these genes have been speculated to control corresponding traits, overexpression of these major genes in common wheat has not been carried out to date, because of the difficulties in transforming wheat.

In this manuscript, *ThMYC4E*, the candidate major gene controlling the blue grain trait from *Th. Ponticum* [31], was transformed into common wheat using *Agrobacterium*. A series of transgenic lines were obtained to evaluate the function of bHLH in regulating anthocyanin biosynthesis and its roles in the formation of the traits related to anthocyanin biosynthesis in common wheat.

## 2. Results

### 2.1. Transgenic Lines Overexpressing ThMYC4E Were Characterized by Colored Tissues

Using *Agrobacterium*-mediated genetic transformation, 18 independent events (T0) were obtained. All transgenic lines carried purple grains, glumes and leaves, while no color change existed in the stems and roots compared with the wild-type (WT) (Figure 1). Because the bHLH transcription factor was inferred to play an important role in the formation of blue and purple grain traits, the grains were transected and the pericarps and aleurone appeared dark (Figure 1A). Unfortunately, purple or blue color was not obvious in the pericarps and aleurone tissues. 

Anthocyanin biosynthesis is reported to be induced by light. For example, white pericarps of immature grains in purple grain wheat cultivars turned purple under light treatment without the glumes [30]. To further investigate the features of anthocyanin biosynthesis in the grain, the glumes were stripped during seed development, 1 week after flowering. The grains without glume protection were always deep purple in color in the transgenic lines, while the grains with glumes did not appearpurple in color until ripe (Figure 1B). No WT grains displayed pigment accumulation over any period under any treatment (Figure 1B).

### 2.2. RNA-Seq and qPCR Indicated that Phenylalanine Biosynthesis and Flavonoid Biosynthesis Were Activated by Overexpression of ThMYC4E

Obvious phenotypes were observed in the transgenic lines, but the influences of the overexpression of *ThMYC4E* were unknown. In order to identify which metabolomic pathway was activated, RNA-Seq analysis was carried out on the grains of the transgenic lines and WT. A total of 375.44 M reads and 56.32 Gb clean bases were obtained with Q20 at 95% after filtering. A total of 181,757 unigenes were obtained and the FPKM value was calculated to evaluate the expression level of each unigene. In total, 920 unigenes were up-regulated in the transgenic lines compared to the WT, while 1550 unigenes were down-regulated (Figure 2A). Among them, 1459 differentially expressed unigenes were enriched in the metabolic pathways related to phenylalanine and flavonoid biosynthesis (Appendix A). Compared with the WT, most of the structural genes associated with the anthocyanin biosynthesis pathway showed higher levels of expression in the transgenic wheat (Figure 2B). Among them, the transcript levels of *F3H, F3’H* and *DFR* were up-regulated by 9.33-, 2.45- and 9.75-fold, respectively (Figure 2B). The overexpression of *ThMYC4E* led to the up-regulation of structural genes related to anthocyanin biosynthesis.

To further investigate the function of *ThMYC4E*, some transcription factors and structural genes were selected to evaluate their transcript levels in several tissues and grains under light treatment using qPCR. *ThMYC4E* showed higher levels of expression in all tissues of the transgenic lines than those of the WT. The expression of *ThMYC4E* was highest in transgenic grains, followed by the roots (Figure 2C). The expression levels of the key structural genes *CHS* and *F3H* in transgenic grains were higher than those in the WT in all tissues except root (Figure 2C). The light treatment of grains had increased the transcript level of *TaMYB7D* significantly (Figure 2C). The expression of *F3′5′H* was higher in the transgenic grains and glumes, while in the leaves, its expression decreased (Figure 2C). The expression of *DFR* was much higher in transgenic grains, while in leaves and glumes its expression decreased compared to the WT (Figure 2C).

### 2.3. Measurements of Chemical Components and Antioxidant Activity 

As mentioned above, the color of the transgenic line tissues was different to the WT and the transcript levels of the structural genes involved in anthocyanin biosynthesis were higher in the transgenic lines than the WT. Although the color of some transgenic line tissues were similar to the WT, the anthocyanin content of all tissues in the transgenic lines were higher than the WT. In the leaves and peeled grains of the transgenic lines, the total anthocyanin contents were nearly 0.722 and 0.510 mg/g, which were highest in these tissues (Figure 3A). Without glumes, light treatment improved the anthocyanins in the grains of the transgenic lines markedly.

To identify the anthocyanin components in the grains, the multiple ion monitoring-enhanced product ions (stepwise MIM–EPI) strategy was used [32]. A total of 592 chemical compounds were detected in the analysis and eight compounds were anthocyanins. They were peonidin 3, 5-diglucoside chloride, peonidin *O*-hexoside,peonidin, rosinidin *O*-hexoside,cyanidin 3,5-*O*-diglucoside (cyanin), peonidin 3-*O*-glucoside chloride,cyanidin 3-*O*-galactoside anddelphinidin 3-*O*-glucoside (mirtillin). Four kinds of anthocyanins, cyanidin 3,5-*O*-diglucoside (cyanin), peonidin 3-*O*-glucoside chloride, cyanidin 3-*O*-galactoside, delphinidin 3-*O*-glucoside (mirtillin), existed only in the transgenic lines (Table 1).

Anthocyanins are suggested to have benefits for human health, partially due to their powerful antioxidant properties and anthocyanin-rich food products have become increasingly popular. Antioxidant activity was evaluated in the grains of the transgenic lines and WT, and those of the transgenic lines showed higher antioxidant activity, in terms of scavenging DPPH radicals, than the WT (Figure 3B).

## 3. Discussion

As wheat transformation is time-consuming, laborious and costly, the overexpression of the functional genes related to anthocyanin biosynthesis in common wheat has not been reported often. This is the first report on the overexpression of the bHLH transcription factors that regulate anthocyanin biosynthesis in common wheat. As expected, based on RNA-seq and qRT-PCR results, the overexpression of *ThMYC4E* induced the transcription of structural genes related to anthocyanin biosynthesis. As a consequence, anthocyanin accumulated in a series of tissues, which resulted in a phenotypic change that was confirmed by measuring the anthocyanin content. Although it could be inferred that *ThMYC4E* induced the transcript of the structural genes and anthocyanin accumulation, it was difficult to determine which gene was key to anthocyanin accumulation in different tissues. The transcript levels of all genes selected in these experiments were not absolutely positively related to the anthocyanin contents in these tissues.

The anthocyanin contents were significantly higher in the roots, stems, grains, leaves and glumes of the transgenic lines than the WT, but only the grain, leaves and glumes of transgenic lines were visually distinct from the WT. Dark grains, purple leaves and purple glumes appeared in the transgenic lines. Interestingly, the grain only appeared darker in color in transgenic lines and not in the WT. The purple grain trait was controlled by two complementary major genes, the bHLH and MYB transcription factors. The expression of bHLH transcription factor alone did not produce the purple grain trait. However, light treatment induced the transcript of *TaMYB7D*. The anthocyanin biosynthesis pathway was activated and the grain became purple in color. The phenotype was similar to the normal purple grain trait found in common wheat [30]. The blue grain trait was speculated to be controlled by one major gene locus and *ThMYC4E* resides in this region. Recently, a trigenic cluster (bHLH-MYB-F3′5′H) was reported to exist in the homologous region to *Ba1* in barley [33]. The trigenic cluster was thought to play an important role in conferring the blue aleurone trait in cereal [33]. This could explain the reason *ThMYC4E* overexpression did not induce the blue grain trait. Further research is therefore required to decipher the genetic code related to the blue grain trait. 

## 4. Materials and Methods

### 4.1. Plant Materials

The wild-type (WT) wheat used in this study was JW1 (*T. aestivum*), provided by Shandong Academy of Agricultural Sciences. Transgenic lines and WT plants were grown in a glasshouse at 25 °C (day)/20 °C (night), with a photoperiod of 16 h light/8 h dark. 

### 4.2. Construction of the Vector and Generation of Transgenic Lines

Specific primers with the adapters, ThMYC4E-pLGY02F and ThMYC4E- pLGY02R, were designed to amplify the target gene *ThMYC4E* from the construct pBRACT214-ThMYC4E [31]. The amplified fragments and PLGY-02 vectors were double-digested with Kpn1 and Spe1, respectively. The procedure was as follows: 2 μL of NEB buffer, 1 U Kpn1, 1 U Spe1, 1 μg of the target fragment/PLGY-01 plasmid were combined and water was added to a final volume of 20 μL. The reaction was carried out overnight at 37 °C. The digested product was subjected to 1% agarose gel electrophoresis. ATIANgel Midi Purification Kit (Tiangen, Beijing, China) was used to obtain a gel-recycled product.

T4 ligase was used with the product to carry outan enzyme-linked reaction to construct a transgene overexpression vector ThMYC4E-PLGY02. The expression vector ThMYC4E-PLGY02 containedthe whole coding sequence of ThMYC4E with a maize ubiquitin promoter. The recombinant vector was then transformed into *Agrobacterium tumefaciens* strain EHA105. Subsequently, immature embryos of JW1 with white grains were infected with the EHA105 carrying the overexpression vector, according to the method described by Zhang et al. [34]. Transgenic lines were selected using specific primers for hyg-F/R and ThMYC4E-F/R, respectively. In further experiments, the T3 family lines carrying the objective gene without the separation were used.

### 4.3. RNA-Seq Analysis

The cDNA libraries of the immature grains 28 days after flowering of the transgenic wheat and JW1 were prepared according to the manufacturer’s instructions for mRNA-Seq sample preparation (Illumina Inc, San Diego, CA, USA). The cDNA library products were sequenced by Illumina paired-end sequencing technology with read lengths of 150 bp using an Illumina HiSeq 2000 instrument (Novogene Bioinformatics Technology Co. Ltd., Beijing, China). The raw sequence reads were stored in the National Center for Biotechnology Information (NCBI) SRA database with the accession number SUB6549510.

Firstly, the low quality reads (>20% of the base qualities were lower than 10), reads with adaptors and reads with unknown bases (N bases more than 5%) were filtered to obtain clean reads. The clean reads were then assembled into Unigenes Trinity software [35]. The gene expression level was evaluated using Fragments Per Kilobase of transcript per Million mapped reads (FPKM). Unigenes that were differentially expressed were analyzed by Chi-square test using IDEG6 software [36]. The false discovery rate (FDR) method was introduced to determine the threshold *p*-value at FDR ≤ 0.001; and the absolute value of |log2Ratio| ≥ 1 was used as the threshold to determine the significance of the differential expression of unigenes. In order to perform functional annotation, the differentially expressed unigenes were submitted to a public database and compared with the Kyoto Encyclopedia of Genes and Genomes (KEGG) databases (http://www.genome.jp/kegg/) using BLASTX (v.2.2.26) with e-values < 1 × e^−5^.

### 4.4. Metabolome Analysis and Measurement of Anthocyanin Content 

The freeze-dried wheat grains were crushed using a mixer mill (MM 400, Retsch, Haan, Germany) with a zirconia bead. One hundred mg of powder were weighed and extracted with 0.6 mL of aqueous methanol (70%) at 4 °C overnight. The extracts were absorbed with CNWBOND Carbon-GCB SPE Cartridge (ANPEL, Shanghai, China) and filtered using anylon syringe filter (SCAA-104, 0.22 μm pore size; ANPEL) after centrifugation.

The sample extracts were analyzed using a liquid chromatography-electrospray ionization-tandem mass spectrometry (LC-ESI-MS/MS) system (HPLC, Shim-pack UFLC CBM30A system Shimadzu, Kyoto, Japan; MS, 6500 Q TRAP Applied Biosystems, Foster city, CA, USA). The analytical conditions were as follows, HPLC: column, ACQUITY UPLC HSS T3 C18 (1.8 µm, 2.1 mm × 100 mm; Waters, Milford, MA, USA); solvent system, water (0.04% acetic acid): acetonitrile (0.04% acetic acid); gradient program,95:5 *v*/*v* at 0 min, 5:95 *v*/*v* at 11.0 min, 5:95 *v*/*v* at 12.0 min, 95:5 *v*/*v* at 12.1 min, 95:5 *v*/*v* at 15.0 min; flow rate, 0.40 mL/min; temperature, 40 °C; injection volume: 2 μL. The effluent was alternatively connected to an ESI-triple quadrupole-linear ion trap (Q TRAP)-MS.

Linear ion trap (LIT) and triple quadrupole (QQQ) scans were acquired on a triple quadrupole-linear ion trap mass spectrometer (Q TRAP), API 6500 Q TRAP LC/MS/MS System equipped with an ESI Turbo Ion-Spray interface, operating in a positive ion mode and controlled by Analyst 1.6 software (AB Sciex, Darmstadt, Germany). The ESI source operation parameters were as follows: ion source, turbo spray; source temperature 500 °C; ion spray voltage (IS) 5500 V; ion source gas I (GSI), gas II(GSII) and curtain gas (CUR) were set to55, 60 and 25.0 psi, respectively; the collision gas (CAD) was high. Instrument tuning and mass calibration were performed with 10 and 100 μmol/L polypropylene glycol solutions in QQQ and LIT modes, respectively. QQQ scans were acquired as multiple reaction monitoring (MRM) experiments with the collision gas (nitrogen) set to 5 psi. Declustering potential (DP) and collision energy (CE)were optimized for individual MRM transitions. A specific set of MRM transitions were monitored according to the metabolites eluted within a set period. The total anthocyanin content was measured according the method described by Ficco et al. [37].

The “Total Monomeric Anthocyanin Pigment Content of Fruit Juices, Beverages, Natural Colorants, and Wines” (AOAC Official Method 2005.02) method was used to measure relative anthocyanin content with three independent experiments. 1%[V/V] HCL was added to 100 mg of comminuted plant tissues (root, steam, leaf, seed and flower) and the mixture was incubated at 4 °C overnight in the dark to extract anthocyanin. Statistical analyses were performed using the software package SPSS for Windows 11.5 (SPSS Inc, Chicago, Illinois, USA) with a 95% confidence interval.

### 4.5. Glume Removal and the Transcription of ThMYC4E and Other Genes in Response to Light

Transgenic lines and WT plants were grown in a glasshouse at 25 °C (day)/20 °C (night), with a photoperiod of 16 h light/8 h dark. One week after anthesis, both the outer and inner glumes were carefully removed from 9 to 10 grains using forceps (light treatment), with the remaining grains in the same spike untreated as controls (dark treatment). Afterwards, the plants were maintained under the same growth conditions. Two days after glume removal, the light-exposed grains and the controls were photographed and then used to investigate the transcriptional response to light with semi-quantitative RT-PCR.

### 4.6. qRT-PCR Analysis

Several peeled and unpeeled grains during the process of maturing were collected at 1 week intervals. Other tissues were also collected: glumes, leaves, stems and roots at grain-filling stage. Specific primers for structural genes were used according to the method described by Jiang et al. [28]. ThMYC4E-RT-F/R and Rc-D1-RT-F/R were designed to evaluate the expression of transcription factors for *ThMYC4E*. Tubulin was used as an internal control gene. All primers arelisted in Appendix A. The transcript levels of these genes were quantified using quantitative real-time PCR (qRT-PCR) with TB Green Premix Ex Taq II (TliRNase H Plus) (Takara Biomedical Technology Co., Ltd., Beijing, China) and an ABI 7500 Real-Time PCR System (Life Technologies, Foster City, CA, USA). The qRT-PCR procedure was as follows: denaturing at 95 °C for 30 s, followed by 40 cycles of 95 °C for 5 s, 60 °C for 34 s; dissociation at 95 °C for 5 s, 60 °C for 1 min and 95 °C for 5 s. The data were analyzed using the 2^−^^ΔΔCT^ method [38]. All biological replicates were measured in triplicate. The primers used in this manuscript are listed in Appendix A. The differences between the transgenic wheat lines and WT were measured by using using Duncan’s multiple range test at 0.05 and 0.01 probability levels in SPSS software version 11.5.

### 4.7. Antioxidant Activity Analysis

The pretreated powder (100 mg) was extracted with 1 mL of 80% ethanol in an ultrasound extractor (500 W, 333 K) for 30 min. The solution was centrifuged for 10 min. DPPH solution was added into ethanol to reach a density of 0.2 mmol/L. The extracted solution was diluted to different concentrations: 100, 50, 25, 12.5, 6.25 and 3.125 mg/mL. Those solutions were added to DPPH at a ratio of 1:1. The mixture reacted for 1 hat room temperature and the absorbance at 510 nm was then measured. The formula was as follows: DPPH radical scavenging activities/% = (1 − Asample/Acontrol) × 100%.

## 5. Conclusions

The basic helix-loop helix (bHLH) transcription factor *ThMYC4E*, the candidate gene controlling the blue grain trait of wheat [31], was transformed to the common wheat variety JW1, and the positive transgenic lines displayed red color in the grains, leaves and glumes, and higher anthocyanin contents in various tissues, and the higher transcript levels of structural genes associated with anthocyanin biosynthesis. Metabolome analysis reveals that four kinds of anthocyanin compounds existed only in the transgenic lines. Moreover, the transgenic lines showed higher antioxidant activity than the WT in terms of scavenging DPPH radicals. This research provides a new wheat germplasm resource with higher anthocyanin content and antioxidant activity, which may be used for the production of functional-foods. 

## Figures and Tables

**Figure 1 ijms-21-00137-f001:**
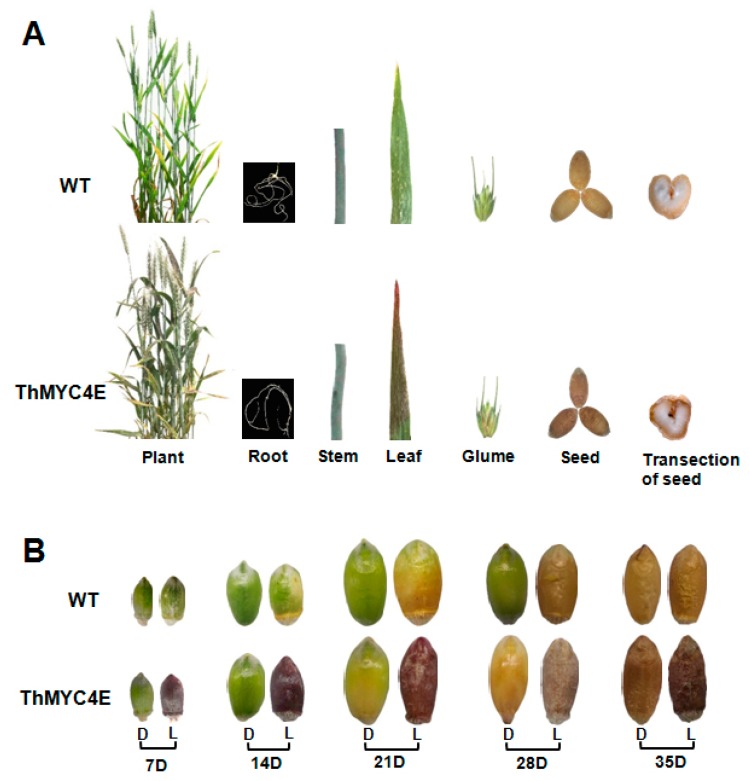
Photograph of transgenic lines and WT under normal conditions and light treatment. (**A**) Photograph of the whole plant and different tissues of transgenic lines and the WT. The transection of seed was form 35 day old fully mature seeds. (**B**) The grains at one week intervals after flowering. 7D = 7 days after flowering, and so on. D = dark treatment (control), while L = light treatment.

**Figure 2 ijms-21-00137-f002:**
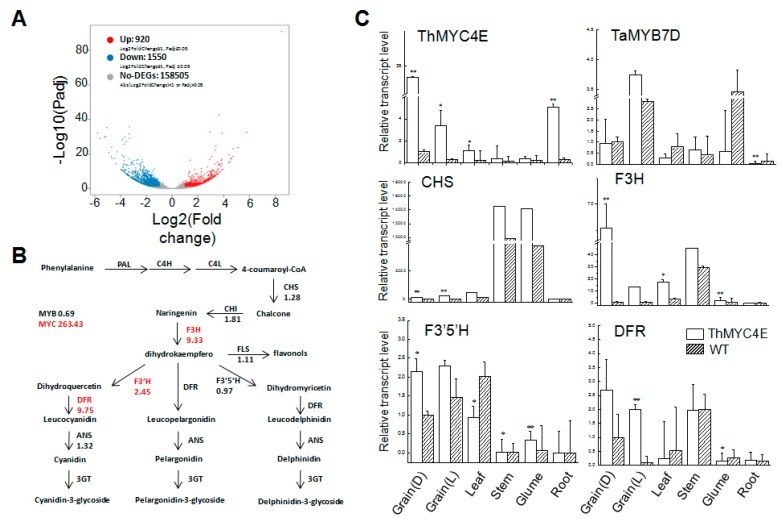
The transcript comparison of the grains in transgenic lines and WT. (**A**) Differentially expressed genes in the grains of transgenic lines and WT as seen following the RNA-seq experiment. The genes were categorized into three classes. Red genes were up-regulated if the gene expression of right-hand sample was larger than the left-hand sample. Blue genes were down-regulated if the gene expression of left-hand sample was larger than the right-hand sample. Grey genes were not differentially expressed. The horizontal coordinate is Log_2_ (fold change) and the vertical coordinate is -Log_10_ (Padj). (**B**) The differences in expression of structural genes in the anthocyanin biosynthesis pathway based on RNA-seq experiment. Arrows show the metabolic stream, the left or upward arrows represent the genes catalyzing the progression of synthesis, the number represents the increased time of expression in the grain of transgenic lines compared with the WT. (**C**) The relative transcript level of the transcription factors and the selected structural genes relative to anthocyanin biosynthesis in different tissues 28 days after flowering. D = dark treatment (control), while L = light treatment. Vertical bars are error bars. All data were compared with the wild type and transgenic lines (* *p* < 0.05 and ** *p* < 0.01).

**Figure 3 ijms-21-00137-f003:**
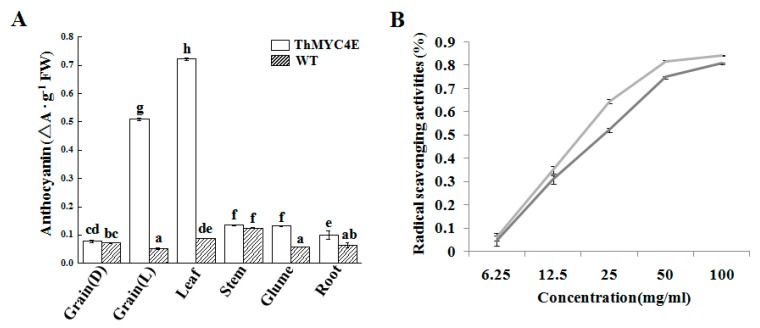
The anthocyanin content of different tissues and DPPH radical scavenging activities in the grains of transgenic lines and the WT. (**A**) The anthocyanin content in different tissues of transgenic lines and the WT. Vertical bars are error bars. Different letters in columns indicate statistically significant differences (*p* < 0.01). (**B**) DPPH radical scavenging activities in the grains of transgenic liens and the WT. X axis represents the concentration of solution and the Y axis represents reducing power. The light gray line represents DPPH radical scavenging activities of transgenic wheat, deep gray line represents DPPH radical scavenging activities of the WT.

**Table 1 ijms-21-00137-t001:** Anthocyanin components and relative contents in the grains of transgenic and WT wheat based on metabolome analysis.

Compounds	Ion Mode	Q1 (Da)	Q3 (Da)	Rt (min)	Molecular Weight (Da)	Ionization Model	The Relative Content of Transgenic Lines	The Relative Contents of WT
Rosinidin *O*-hexoside	Positive	477.1	315	3.31	477.1	Protonated	175,000	63,400
Peonidin *O*-hexoside	Positive	463.1	301	3	463.123	Protonated	105000	61400
Peonidin	Positive	301.1	286	3.98	301.1	Protonated	81,500	46,400
Cyanidin 3,5-*O*-diglucoside	Positive	611	287	2.15	611	Protonated	75,200	9
Cyanidin 3-*O*-galactoside	Positive	449.1	286.8	2.68	448.101	[M]+	195,000	9
Peonidin 3-*O*-glucoside chloride	Positive	463.1	301.1	2.94	498.093	[M-Cl]+	129,000	9
Peonidin 3, 5-diglucoside chloride	Positive	625.4	301	2.16	625.4	Protonated	5380	13,300
Delphinidin 3-*O*-glucoside (Mirtillin)	Positive	465.1	303	2.38	465.1	Protonated	608,000	9

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
