# Peer review of "Overexpression of ThMYC4E Enhances Anthocyanin Biosynthesis in Common Wheat"

_ijms, 2019, doi:10.3390/ijms21010137_

Round 1

Reviewer 1 Report

In the manuscript “Overexpression of ThMYC4E enhances anthocyanin 3 biosynthesis in common wheat”, the authors report about the effect of ThMYC4E transferred into wheat on the color of tissues, level of biochemical compounds and gene expression. For the first time, the role of transcriptional factor ThMYC4E in the regulation of anthocyanin synthesis was demonstrated. Its higher antioxidant activity may be useful in the development of functional food.

However, there are several comments and revisions that should be addressed.

Lines 69-70 “The bHLH transcription factor ThMYC4E has been inferred…” – was it found in Thinopyrum ponticum as described in [30]? I suggest that you mention it. Otherwise, it looks like it was found in the wheat genome.

Lines 19-20 “ThMYC4E, the candidate gene 20 controlling the blue grain trait, was transferred to the common wheat JW1” – Please, clarify, from what was ThMYC4E transferred?

Lines 73-74 “In this manuscript, ThMYC4E, the candidate major gene controlling the blue grain trait, was transformed into common wheat using Agrobacterium” Please, clarify what was the donor of ThMYC4E?

Lines 198-199 “Specific primers with the adapters, ThMYC4E-pLGY02F and ThMYC4E- pLGY02R, were designed to amplify the target gene ThMYC4E” What organism was used as a source for DNA template to amplify the target gene?

Figure 1. The resolution of Figure 1 seems to be low. Perhaps, you should provide a picture of a higher resolution for publication.

Figure 2C What do the vertical bars on diagram charts stand for? Error bars, standard deviation, confidence intervals? Please, specify in the Figure caption and add a paragraph/sentence about statistical analysis to “qPCR analysis” section.

Figure 2C What do the letters on the diagram charts stand for? Please, specify in the Figure caption. If they designate grouping by the significance of the differences, then why in F3H Grain (D) transgenic is “a” and Grain (D) WT is “abc”. It would mean that they do not differ (the common letter is “a”). But the difference between them is more than 60-fold. I cannot understand it.

Figure 2C-CHS Please, add vertical bars to this diagram. If they are no visible at such scale, it is worth making a table with the data obtained from qPCR assay. Alternatively, it is enough to demonstrate the differences between WT and transgenic plants in folds (by numbers) above the charts to easily compare them.

Figure 2C, Figure 3 Please, decode Grain (D) and Grain (L) in the Figure caption.

Lines 116-117 “The expression levels of the key structural genes CHS and F3H in transgenic grains were higher than those in the WT in all tissues (Figure 2C).”  I cannot see in the diagrams CHS and F3H that the expression level of these genes is higher in transgenic plants compared to WT in all tissues in grains. Did you measure different tissues in grains? Perhaps, you meant “in transgenic plants”? If so, the differences between WT and transgenic plants are not obvious from the diagrams especially for root (in both cases they have letters “c”).

Line 280 “Tubulin was used as an internal control gene.” In fact, it is not enough to use one internal control for confident results of qPCR. It would be better if the authors confirmed their results using an alternative control gene. However, if the authors demonstrate the fold-differences between WT and transgenic plants in the diagrams in Figure 3C, it will be okay.

“qPCR analysis” What method of statistical analysis did you perform to reveal if there are significant differences between data? Did you measure the efficiency of the primers?

There seem to be problems with text formatting since some spaces between words disappeared. Here what I found. Please, carefully revise the text and correct all such misprints.

Line 38 “red to purple andblue in flowers” – “red to purple and blue in flowers”

Line 45 “benefits tohuman health” – “benefits to human health”

Line 47 “lengthin model plants” – “length in model plants”

Line 55 “and critical rolesin” – “and critical roles in”

Line 74 “Aseries of transgenic lineswere obtained” – “A series of transgenic lines were obtained”

Line 75 “itsroles in the formation” – “its roles in the formation”

Line 81 “compared with the WT” – Please, decipher “WT”

Line 85 “biosynthesis is reported tobe induced by” – “biosynthesis is reported to be induced by”

Line 101 “A totalof” – “A total of”

Line 126 “Green genes weredown-regulated” – “Green genes were down-regulated”

Line 126 “Green genes weredown-regulated” – In Figure 2A they look blue

Line 125 “Red genes wereup-regulated” – “Red genes were up-regulated”

Line 127 “Bluegenes werenot” – “Blue genes werenot”

Line 127 “Bluegenes werenot” – In Figure 2A they look grey

Line 130 “theleft or upward” – “the left or upward”

Line 132 “compared with theWT” – “compared with the WT”

Line 178 “grain, leaves and glumesof transgenic” – “grain, leaves and glumes of transgenic”

We recommend the manuscript for publication after the listed issues will be corrected or addressed.

Kind regards,

Reviewer.

Reviewer 2 Report

The article is very interesting and timely. I have a few suggestions if authors could address those during the revision.

1) In the introduction, it will be great if authors can add more information about the natural variation for anthocyanin content in the wheat and what are the doner lines are being used worldwide to get high anthocyanin wheat.

2) The author needs to provide justification for why overexpression is important. Is this experiment is to enhance our understanding or looking to develop a product? 

3) In results, "18 positive lines were obtained" are those total number of transgenic plants (T0) or independent events. How many independent events authors studied. 

4) In Figure 1. is that transection of seed is form 35 day old fully mature seeds?

5) Why CSH does not have an error bar, in fig 2

6) In the conclusion "This research provides a new wheat germplasm resource with higher anthocyanin content and antioxidant activity, which may be used for the production of functional-foods" - what are the advantages of novelty over the naturally occurring black and purple wheat germplasm. 
